# The Genetic Evolution of DENV2 in the French Territories of the Americas: A Retrospective Study from the 2000s to the 2024 Epidemic, Including a Comparison of Amino Acid Changes with Vaccine Strains

**DOI:** 10.3390/vaccines13030264

**Published:** 2025-03-01

**Authors:** Alisé Lagrave, Antoine Enfissi, Sourakhata Tirera, Magalie Pierre Demar, Jean Jaonasoa, Jean-François Carod, Tsiriniaina Ramavoson, Tiphanie Succo, Luisiane Carvalho, Sophie Devos, Frédérique Dorleans, Lucie Leon, Alain Berlioz-Arthaud, Didier Musso, Raphaëlle Klitting, Xavier de Lamballerie, Anne Lavergne, Dominique Rousset

**Affiliations:** 1Arbovirus National Reference Center, Virology Unit, Institut Pasteur de la Guyane, Cayenne, French Guiana; alagrave@pasteur-cayenne.fr (A.L.); aenfissi@pasteur-cayenne.fr (A.E.); stirera@pasteur-cayenne.fr (S.T.); anne.lavergne@pasteur.fr (A.L.); 2Laboratoire Centre Hospitalier de Cayenne, Cayenne, French Guiana; magalie.demar@ch-cayenne.fr (M.P.D.); jean.jaonasoa@ch-cayenne.fr (J.J.); 3Department of Biology, West French Guiana Hospital Center, Saint-Laurent-du-Maroni, French Guiana; jf.carod@ch-ouestguyane.fr (J.-F.C.); t.ramavoson@ch-ouestguyane.fr (T.R.); 4Santé Publique France, Cellule Guyane, Cayenne, French Guiana; tiphanie.succo@santepubliquefrance.fr (T.S.); luisiane.carvalho@santepubliquefrance.fr (L.C.); sophie.devos@santepubliquefrance.fr (S.D.); 5Santé Publique France, Cellule Antilles, French Caribbean Islands; frederique.dorleans@santepubliquefrance.fr (F.D.); lucie.leon@santepubliquefrance.fr (L.L.); 6Laboratoires Eurofins Guyane, French Guiana; alain.berlioz@biologie.eurofinseu.com (A.B.-A.); didier.musso@biologie.eurofinseu.com (D.M.); 7National Reference Center for Arboviruses, Inserm-IRBA, Marseille, France; raphaelle.klitting@inserm.fr (R.K.); xavier.de-lamballerie@univ-amu.fr (X.d.L.); 8Unité des Virus Émergents (UVE: Aix-Marseille Univ, Università di Corsica, IRD 190, Inserm 1207, IRBA), Marseille, France

**Keywords:** DENV2, French territories of the Americas, epidemic, genomic diversity, phylogenetic analysis, amino acid variations, vaccines, prevention strategies

## Abstract

Background: Dengue virus type 2 (DENV2) is endemic to hyperendemic in the French territories of the Americas (FTAs), including French Guiana, Guadeloupe, Martinique, Saint-Barthelemy, and Saint-Martin. In 2023–2024, French Guiana, Martinique, and Guadeloupe experienced unprecedented dengue epidemics partly associated with this serotype. In response, we conducted a retrospective study of the diversity of DENV2 strains circulating in the FTAs from 2000 to 2024. Methods: To this end, we selected DENV2 samples from the collection at the National Research Center for Arboviruses in French Guiana (NRCA-FG) and sequenced them using Oxford Nanopore Technologies (ONT)-based next-generation sequencing (NGS). Results: Phylogenetic analysis revealed that (i) the 77 DENV2 sequences from the FTAs belong to two distinct genotypes—Asian American and Cosmopolitan; (ii) from the 2000s up to the 2019 epidemic in French Guiana, all sequenced strains belonged to the Asian American genotype; (iii) and from 2019 to 2020, strains circulating in Martinique and Guadeloupe belonged to the Cosmopolitan genotype, specifically the Indian subcontinent sublineage, while (iv) strains from the 2023–2024 outbreak in Martinique, Guadeloupe, and French Guiana fall within a distinct sublineage of the same genotype—Other Cosmopolitan. Additionally, we analyzed amino acid (AA) changes in FTA sequences compared to the Dengvaxia^®^ and Qdenga^®^ vaccines. The analysis of amino acid changes in FTA sequences compared to the vaccines (Dengvaxia^®^ and Qdenga^®^) identified 42 amino acid changes in the prM/E regions (15 in the prM region and 27 in the E region) relative to CYD-2 Dengvaxia^®^ and 46 amino acid changes in the prM/E regions relative to Qdenga^®^, including 16 in the prM region and 30 in the E region. Some of these AA changes are shared across multiple genotypes and sublineages, with 8 substitutions in the prM region and 18 in the E region appearing in both analyses. This raises questions about the potential impact of these changes on vaccine efficacy. Conclusion: Overall, these findings provide a current overview of the genomic evolution of DENV2 in the FTA, which is crucial for developing more effective prevention and control strategies and for selecting future vaccines tailored to circulating strains.

## 1. Introduction

Dengue virus (DENV), a member of the *Flaviviridae* family, is the most common arthropod-borne acute febrile illness. The DENV genome is a 10,700-base single-stranded positive RNA encoding structural (capsid (C), pre-membrane (prM), and envelope (E)) and non-structural proteins (NS1, NS2A, NS2B, NS3, NS4A, NS4B, and NS5) [1]. The outer surface of the viral structure is made up of structural proteins that facilitate viral entry into human cells, while non-structural proteins are associated with viral replication [2]. Four distinct serotypes (DENV1, DENV2, DENV3, DENV4) are defined by their genetic characteristics (i.e., 25% genetic differences between two serotypes) [3]. Within these serotypes, different genotypes co-exist, separated by a 6% genetic difference [4]. Amino acid sequence heterogeneity between serotypes is around 40%, and the divergence generated within serotypes is around less than 9% [2].

Dengue is endemic in tropical and subtropical regions. In November 2024, more than 14 million cases of dengue fever have been reported globally, including more than 10,000 deaths [5]. While a substantial increase in dengue cases has been reported worldwide over the past five years, this increase has been particularly pronounced in the region of the Americas, where the number of cases exceeded 12 million in November 2024, surpassing the annual peak of 4.6 million cases in 2023 [6]. In the FTAs, dengue circulation is endemo-epidemic to hyper-endemic (regular circulation with epidemic peaks and co-circulation of different serotypes). The various epidemics between 2005 and 2021 have been described in Bonifay et al., 2023 [7]. DENV2 caused epidemic outbreaks in French Guiana in 2006, co-circulated with DENV4 in 2012–2013, with DENV1 in 2020–2021, and with DENV3 in 2023–2024 [7,8]. In Martinique, DENV2 was the predominant serotype during the 2013–2014 epidemic and during the 2023–2024 outbreak [9,10]. In Guadeloupe, DENV2 circulated predominantly during the 2019–2021 outbreak and in the same way during the 2023–2024 outbreak [9,10]. The different genotypes of the DENV2 include one sylvatic genotype and five non-sylvatic genotypes (Asian I, Asian II, American, Asian-American, and Cosmopolitan) [11]. The sylvatic and American genotypes no longer circulate, the latter having been over-circulated in Central and South America in the 1950s and 1960s [12]. Asian I and II genotypes are specific to the Asian continent [13]. The Asian-American genotype has been spreading in Southeast Asia and in Central and South America since the 1960s, but has not been detected in Asia since the 2000s [12]. The last one, the Cosmopolitan genotype, is geographically widespread and contributes the most to the current global burden of DENV2 [13,14,15].

There are only a limited number (n = 25) of DENV2 sequences from French Guiana in public databases, and no study has examined the DENV2 strains circulating in the FTAs from 2000 to 2024. Herein, we conducted a retrospective analysis of the diversity of DENV2 strains in the region over this period. We also considered the potential impact of this diversity by assessing amino acid (AA) changes in the sequences from French Guiana in comparison to the Dengvaxia^®^ and Qdenga^®^ vaccines.

## 2. Materials and Methods

### 2.1. Ethical Statement

This study was conducted as part of the public health surveillance program of the National Reference Center for Arboviruses in French Guiana (NRCA-FG), in collaboration with the French Public Health Agency (Santé Publique France, SPF). Accordingly, as an epidemiological record, consultation with the ethics committee was not required. Samples involved in this study were chosen among remaining human serum samples received as part of standard diagnostic and expertise activities of the NRCA-FG and kept by the NRCA-FG biobank according to the French legislation (article L.1211–2 and related of the French Public Health Code—FPHC). Human serum samples were anonymized, with no or minimal risk to patients within the terms of the European General Data Protection Regulation and the French National Commission on Informatics and Liberty (CNIL).

### 2.2. Clinical Samples and Study Design

The samples used in this study came from the NRCA-FG biobank, which comprises clinical specimens received from hospital and private laboratories in French Guiana and French Caribbean Islands as part of the arbovirus surveillance system. No information was available on the evolution of the cases (severity). Between 2001 and January 2024, 3616 samples tested DENV-2-positive, among which 624 had a cycle threshold (Ct) value below 26. Among these samples, a selection was made in each municipality and for each detection period, leading to 77 samples collected between 2001 and 2024 being selected for the analysis (Table 1), including 12 from Guadeloupe, 5 from Martinique, 3 from Saint-Barthelemy, 3 from Saint-Martin, 51 from French Guiana, and 3 from French Guiana imported from French Caribbean Islands.

### 2.3. MinION Library Preparation and Multiplexed Nanopore Sequencing

Whole-genome sequencing was performed on an Oxford Nanopore MinION device using R9.4 flow cells (Oxford Nanopore Technologies, Oxford, UK) based on a protocol from Quick et al., 2017 [16]. RNAs were first extracted using the Macherey Nagel Virus 96 kit (Macherey Nagel, Düren, Germany; ref: 740452.4) according to the manufacturer’s instructions. Then, 8 μL of RNA was reverse transcribed to complimentary DNA using LunaScript RT SuperMix (NEB, Ipswich, MA, USA), and a serotype-specific multiplex PCR was performed with Q5 High-Fidelity DNA polymerase (NEB, Ipswich, MA, USA) in three separate pools, using a DENV2 primer scheme described in Appendix B Table A1. Primer schemes were designed using the modified CDC protocol [17,18,19] and empirically refined. Cycling conditions for the RT were 25 °C 2 min 1 cycle, followed by 55 °C 10 min 1 cycle and 95 °C 1 min 1 cycle. For the multiplex PCR, the cycling conditions were 30 s at 98 °C, followed by 35 cycles at 98 °C for 15 s, 63 °C (Pool 1 and 2) or 60 °C (Pool 3) for 5 min, and conservation at 4 °C. The resulting 400 bp PCR products were pooled, cleaned using AmpureXP magnetic beads (Beckman Coulter, High Wycombe, UK), and quantified using a Qubit dsDNA High Sensitivity assay on a Qubit 3.0 instrument (Thermo Fisher Scientific, Waltham, MA, USA). The samples were then barcoded using the Ultra II End Repair/dA-Tailing Module (New England Biolabs, Ipswich, MA, USA) and the native barcoding kits NBD104 and NBD196 (Oxford Nanopore Technologies), cleaned with magnetic beads, and pooled at equimolar ratios before ligation of the AMII adapters with blunt/TA ligase master mix (New England Biolabs, Ipswich, MA, USA). Sequencing libraries were loaded onto the R9.4 flow cell using the ligation sequencing kit SQK-LSK109 (Oxford Nanopore Technologies, Oxford, UK), and sequencing data were collected for 12 h. Sequence reads were base-called and demultiplexed using the Guppy algorithm v3.6 (Oxford Nanopore Technologies, Oxford, UK). The consensus genome sequences were produced using the Artic network’s bioinformatics pipeline (https://artic.network/ncov-2019/ncov2019-bioinformatics-sop.html, accessed on the 17 July 2024), along the primer schemes used to produce amplicons, which read alignment to genomes of reference with minimap2, incorporating primer removal and the nanopolish algorithm (https://github.com/jts/nanopolish, accessed on the 17 July 2024) to improve the consensus sequence by signal level variant calling. Regions with insufficient coverage (minimum depth coverage set to 20) were masked with N characters.

### 2.4. Phylogenetic Analysis

The 77 whole-genome consensus sequences of DENV2 obtained in this study (Appendix B Table A2) have been aligned with CLC Main Workbench software (version 22; Qiagen, Hilden, Germany) to whole-genome DENV2 references sequences downloaded from GenBank (28 February 2024). A total of one hundred and twenty-two reference sequences were selected to ensure a balanced geographical and temporal distribution, encompassing all the genotypes described so far. For this, using GISAID, representatives of each genotype and sub-genotype were chosen to reflect the different isolation dates and localities. In a second step, the NRCA-FG generated sequences were subjected to BLAST (https://blast.ncbi.nlm.nih.gov/Blast.cgi?PROGRAM=blastn&BLAST_SPEC=GeoBlast&PAGE_TYPE=BlastSearch, accessed on the 20 July 2024), and the closest reference sequences for each were added to the dataset, resulting in a final dataset of 122 reference sequences (Appendix B Table A3). BEAST and BEAUTI (version 1.10.1) software were used to construct the maximum clade credibility (MCC) tree, using GTR + G + I (General Time Reversible with Gamma distribution and Invariant sites) substitution model, corresponding to the best-fit tested on CLC Main Workbench software (version 8.0), according to the corrected Akaike Information Criterion (AICc), with a strict clock model and Bayesian skyline prior [20,21,22]. The Markov Chain Monte Carlo (MCMC) analysis was run for 50 million generations with sampling every 1000 generations. The final MCMC sampling chains were checked using Tracer v1.7.1 with 10% burn-ins removed. Convergence was controlled by checking that an effective sample size (ESS) > 200 was obtained for all parameters using Tracer v1.7.1 [23]. The maximum clade credibility tree was generated by Tree Annotator 1.10.4, and the time-scaled phylogeny was visualized using FigTree v1.4.3 (http://tree.bio.ed.ac.uk/software/figtree/, accessed on the 20 September 2024).

### 2.5. Amino Acid Changes Against Vaccinal Strains Dengvaxia and Qdenga

Amino acid changes were compared between the 77 FTA sequences generated in this study plus 17 reference sequences from Martinique and Guadeloupe (OR229958 to OR229960; OR229962 to OR229973; OR229975 to OR229976) and the Dengvaxia CYD-2 vaccine strain (Accession number: KX239895) and Qdenga vaccine strain (Accession number: U87411). A Python script was written to generate a tab-separated file with the position where amino acid changes occurred and the corresponding alternative amino acid, for each reference (Dengvaxia CYD-2 vaccine strain and Qdenga vaccine strain). The vaccine composition of Dengvaxia CYD-2 vaccine strain (KX239895) includes the prM (membrane glycoprotein precursor) and E (envelope) regions, while Qdenga vaccine strain (U87411) includes all genome regions (C, prM, E, NS1, NS2A, NS2B, NS3, NS4A, NS4B, NS5). We focused only on the common regions (prM and E), as the majority of neutralizing antibodies target the viral E proteins [24,25].

## 3. Results

The global phylogenetic analysis was carried out on 199 sequences (Figure 1). FTA DENV2 sequences obtained from French Guiana (FG), Martinique (MTQ), Guadeloupe (GLP), Saint-Martin (SM), and Saint-Barthelemy (SB), having circulated between 2002 and 2024, belong to different genotypes—Asian American and Cosmopolitan (Figure 1).

### 3.1. DENV2 Strains Belonging to the Asian American Genotype

The Asian American genotype includes FTA strains from 2007 through the 2019 FG outbreak, as shown in Figure 2. Strains from French Guiana circulating in 2002 (EU920835) and 2006–2007 (EU920848 and R0113041) are intermingled with other American strains (Guyana, Suriname, Dominican Republic, Cuba, or Nicaragua). Strains from Martinique 2005 (EU9208037, EU9208043) and Guadeloupe 2006–2007 (EU9208049 and R0124144) appear in a different lineage, with the most recent common ancestor (MRCA) dating back to 2004 (95% HPD 2004–2005). Similarly, the Guadeloupe 2012 (W1221159) and French Guiana 2013 (PP719122-PP719125) strains appear in a different sub-lineage with a common ancestor dating from 2009 (95% HPD 2008–2011). The sequences from FG 2019 (PP719127 to PP719138) are also found in a distinct sublineage with a common ancestor dating from 2018 (95% HPD 2018–2019). These sequences are found in a monophyletic clade sharing nucleotide identity from 99.88 to 100% between each other and from 99.41 to 99.49% nucleotide identity with a sequence from Brazil PP234938 Brazil circulating in 2018.

### 3.2. DENV2 Strains Belonging to the Cosmopolitan Genotype

Interestingly, at the same time (2019–2020), strains circulating in Martinique and Guadeloupe (2019–2020 epidemic) belong to another genotype (Cosmopolitan genotype, Indian sub-continent lineage), as shown in Figure 3. The sequences of this monophyletic clade are distributed according to their origin (MTQ or GLP) and share between 99.60 and 99.96% nucleotide identity with each other and 99.55 and 99.72% nucleotide identity with MN272404 Reunion Island 2018. The MRCA between MN272404 Reunion Island 2018 and this clade dates to 2015 (95% HPD 2014–2017), Figure 3. The Reunion Island strain was imported from the Seychelles [27]. Asian strains, notably Indian strains, are at the origin of this clade.

Finally, the sequences of the 2023–2024 DENV2 epidemics in the FTAs, mainly in French Guiana, Guadeloupe, and Martinique, but also of a few sporadic cases in Saint-Martin and Saint-Barthelemy, belong to a well-supported monophyletic clade (posterior probability 100%) and are included in the Cosmopolitan genotype, the Other Cosmopolitan sublineage. These sequences share 99.78–100% nucleotide identity. In addition, sequences from this clade shared 99.11–99.30% nucleotide identity with Indian sequences from 2021 (OM639992 and OM639982). The MRCA between those sequences’ dates back to 2016 (95% HPD 2014–2017), as shown in Figure 3.

### 3.3. Amino Acid Changes Against the Vaccines (Dengvaxia and Qdenga)

Comparison of amino acid variations between our dataset containing the 77 FTA sequences generated in this study added to the 17 sequences from Martinique and Guadeloupe 2019–2020 and the Dengvaxia CYD-2 vaccine strain (KX239895) showed 42 positions with AA substitutions, including 15 in the prM (membrane glycoprotein precursor) region and 27 in the E (envelope) region (Appendix A). The ratio between the number of substitutions and the AA size of each region, respectively, 166 AA for prM and 495 AA for E, shows a higher ratio for prM (9.04%) compared to E (5.45%). A closer look at the AA substitution specific to one or more genotypes and subgenotypes (Asian American, Cosmopolitan Indian sub-continent, and Cosmopolitan Other Cosmopolitan) compared with the Dengvaxia CYD-2 vaccine strain (KX239895) reveals 10 positions with AA variations in the prM region (ratio 6.02%) and 19 in the E region (ratio 3.83%), as shown in Figure 4 and Appendix A. Some are shared by all sequences of the Asian American genotype (n = 17), of the Cosmopolitan Indian subcontinent subgenotype (n = 14), or of the Other Cosmopolitan subgenotype (n = 17), as shown in Figure 4. Some are even shared between all genotypes (n = 5), as shown in Figure 4. It is therefore possible to define genotype-specific substitution profiles.

Analysis of amino acid changes between our dataset compared to the Qdenga vaccine strain (U87411) showed 46 AA positions with AA substitutions along the prM and E regions with, respectively, 16 and 30 AA changes, as shown in Appendix A. Looking more specifically at the AA substitution specific to one or more genotypes and subgenotypes (Asian American, Cosmopolitan Indian sub-continent, and Cosmopolitan Other Cosmopolitan) compared to the Qdenga vaccine strain, we observed 32 amino acid changes with 11 in the prM (ratio of 6.62%) and 21 in the E (ratio of 4.24%), as shown in Figure 5 and Appendix A. Some AA substitutions are even shared between all genotypes as for the Asian American genotype (n = 17) and for the Cosmopolitan genotype (n = 21), as shown in Figure 5. Within this Cosmopolitan genotype, the Cosmopolitan Indian subcontinent shows 20 shared AA changes, while the Other Cosmopolitan subgenotype shows 21, as shown in Figure 5. Some are shared by all genotypes (n = 8), as shown in Figure 5. It is therefore possible to define genotype-specific substitution profiles.

Moreover, some amino acid changes specific to one or more genotypes and subgenotypes are common to both analyses (Dengvaxia and Qdenga vaccine strains) in the prM (8 common amino acid changes) and E (18 common amino acid changes) regions, as shown in Table 2 and Figure 4, Figure 5 and Appendix A.

## 4. Discussion

This study describes the evolution of DENV2 strains circulating in the FTAs over a 20-year period and the corresponding amino acid variations compared to vaccine strains (Dengvaxia^®^ and Qdenga^®^). The phylogenetic analysis of 100 DENV2 sequences circulating in the FTAs, between 2002 and 2024, reveals the successive circulation of two distinct genotypes: Asian American and Cosmopolitan. The first one included all strains from 2000s to 2013, as well as strains circulating in 2019 in French Guiana (mainly affecting the municipality of Kourou). FG sequences are intermingled with other sequences from America, suggesting that significant exchanges occurred in the region during this period [28,29,30]. At the same time, in 2019–2020, strains from Martinique and Guadeloupe were shown to belong to the Cosmopolitan genotype under the Indian sub-continent lineage (or 2II_A according to Hill et al., 2024 [26]). The exact importation origin could not be determined, but these sequences presented high homology with sequences from the Indian Ocean (Reunion Island and the Seychelles) and India, which circulated between 2016 and 2018 [9]. This sub-lineage seems to have already circulated in the Americas (such as OR389333 Venezuela 1990), but its origin seems to be Asian (India) [31,32]. Finally, French Guiana, Guadeloupe, Martinique, Saint-Martin, and Saint-Barthelemy strains from the 2023–2024 epidemic were found to belong to the Cosmopolitan genotype, lineage Other Cosmopolitan (or 2II_F according to Hill et al., 2024 [26]). Other American strains over the same period belong to this lineage such as OM791800 and OM791801 Peru 2019, ON634745 and OP941843 Brazil 2022; and OR037352 Colombia 2022. These sequences are related to Bangladesh strains from 2017 and 2019 (LC436673 and MN328061) [9,33,34,35]. However, the 2023–2024 FTA outbreak clade is distinct from this one; therefore, new introductions from Asia (MRCA 2016) can be assumed to be the origin of this epidemic instead of a drift of strains already present on the American continent. The 2023–2024 FTA outbreak clade found the highest percentage of nucleotide identity with the Indian sequences of 2021 (OM639992 and OM639982) in GenBank. However, other routes of introduction cannot be excluded, as these data depend on what is available in databases, which are not exhaustive.

The brief interval between the 2019–2021 and 2023–2024 outbreaks of the same serotype (DENV2) in French Guiana and Guadeloupe was unexpected, as it contrasts with the typical historical pattern, where the predominant serotypes generally alternate between two consecutive outbreaks, with an inter-epidemic period of 3–4 years. This raises the question of what factors have driven the emergence and spread of this serotype. Different mechanisms could be at play. While population immunity evolves over time due to births and population turnover, a loss of immunity alone does not explain the rapid spread observed over such a short period (from 2019–2021 to 2023–2024). Other factors must also contribute to the spread and diversification of DENV, such as environmental and socioeconomic factors that facilitate vector adaptation and proliferation [29,36,37,38,39,40,41,42,43,44]. A transmission advantage due to vector-related factors could be involved, but no entomological data related to vector competence and more broadly to vectorial capacity are available. A transmission advantage due to higher viremia is difficult to evaluate because of possible sampling biases, but, based on our data, no significant difference in DENV2 Ct values was observed between the 2019–2020 and 2023–2024 epidemics in French Guiana. Finally, the genetic variability induced by the genotype shift in French Guiana and subgenotype shift in Guadeloupe could have led to at least partial escape from population immunity. Thus, by comparing the amino acid changes in the envelope region of the 2023–2024 epidemic strains from French Guiana and Guadeloupe (Cosmopolitan Other Cosmopolitan) with those from the 2019 epidemic in French Guiana (Asian American genotype) and 2019–2020 in Guadeloupe (Cosmopolitan Indian Subcontinent genotype), variations are observed at positions V23I, H52Q, V91I, V141I, N149H, I170T, N203D, V308I, V322I, M340T, S363A, I380V, S390N, A447V, V462I, and V491A for French Guiana and V23I, H52Q, I141V, and N149S for Guadeloupe. Some of these amino acid changes are found in known epitopes of the DENV2 envelope, such as V308I, M340T, S363A, and S390N (IEDB Database). These AA changes could affect these specific epitopes and may impact the immune response against the virus. Therefore, these changes could contribute to this escape. The significant and increasing diversity observed has led the scientific community to reconsider the classification of DENV viruses in order to establish a more precise and accurate naming system [26]. In the future, it would be valuable to explore the relationship between genomics and clinical data, aiming to identify variants that may be associated with specific clinical data (e.g., severity).

The genetic diversity of DENV raises the question of its impact on antigenic characteristics, and therefore on infection- or vaccine-elicited antibodies, and this poses a major challenge for obtaining an effective vaccine [45]. Vaccine development against dengue has progressed during the last decade [46]. However, these efforts have been challenged by the need to induce effective protection against all four DENV serotypes and by the potential negative effect of partial immunity to DENV, which can worsen symptoms in subsequent heterotypic infection [46]. At present, the most advanced dengue vaccines are all tetravalent and based on live-attenuated recombinant viruses. CYD-TDV, Dengvaxia^®^ developed by Sanofi Pasteur (Lyon, France), and TAK-003, DENVax, Qdenga^®^ developed by Takeda (Cambridge, MA, USA), have been approved, although Dengvaxia^®^ will no longer be marketed from August 2026. The original strain of Dengvaxia^®^ is the Thailand DENV-2 PUO-218 isolated in 1980, while DENV-2 Thailand 16681 used in Qdenga^®^ was isolated in 1984; both are of the Asian I genotype [47,48]. Consequently, these vaccine strains are genetically distant from the Cosmopolitan genotype, which is responsible for the most recent global dengue epidemics.

The amino acid variations between the FTA DENV2 strains and the two vaccines were analyzed. The results identified 42 positions with amino acid changes (15 in the prM region and 27 in the E region) compared to CYD-2 Dengvaxia^®^, while 46 variable amino acid positions were found relative to Qdenga^®^ in the same regions, including 16 in the prM region and 30 in the E region. Some amino acid changes were common across multiple genotypes and subgenotypes, appearing in both analyses (Dengvaxia^®^ and Qdenga^®^), with 8 substitutions in the prM region and 18 in the E region. Among the positions showing conserved amino acid changes relative to the two vaccine strains, nine (Q52H, V91I, I141V, H149N, I164V, N203D, V308I, I322V and N390S) were also previously identified by Martinez et al. from a panel of DENV2 representative of global genetic diversity chosen to investigate the influence of genetic diversity on vaccine performance [48]. Martinez et al. identified several highly variable amino acid positions in the E protein of DENV2 (at positions 71, 129, 141, and 390) and demonstrated that these genetic variations have an impact on the extent of the neutralizing antibody activity following infection and vaccination [47]. Rabaa et al. and Velasco et al. identified significant amino acid changes and mismatches at mAb-targeted epitope sites (E71, E149, and E226) between circulating DENV-2 viruses and vaccine components, including two amino acid changes found in our study (at positions 71 and 149) [49,50]. To further investigate the impact of amino acid modifications on vaccine efficacy, it would be interesting to assess the level of neutralization efficiency of post-vaccination immune sera against these different strains using the PRNT method. Thus, the diversity in the E protein could have several significant implications. (i) Escape from neutralizing antibodies: Alterations in these amino acid positions may modify the E protein’s structure, making it harder for neutralizing antibodies to recognize and bind to the virus. This could impair the immune system’s ability to effectively neutralize the virus. (ii) Immune pressure and evolution: The high variability at these positions suggests they are under immune pressure, likely due to prior host exposure to the virus. Variants with these mutations could have a selective advantage, as they are less susceptible to neutralizing antibodies generated by previous infections or vaccinations. (iii) Impact on vaccine development: The observed variability complicates the development of a universal vaccine. A vaccine targeting several genotypic variants may be needed to develop a universal vaccine [51,52,53,54,55].

## 5. Conclusions

In conclusion: the findings emphasize the significant regional and international exchanges that have led to multiple introductions of DENV2 into the FTAs, particularly from Asia, resulting in epidemic outbreaks. They underscore the importance of genomic research for understanding the evolutionary history of circulating strains and for tracing importation events. The analysis of amino acid changes between FTA strains and vaccine strains (Dengvaxia^®^ and Qdenga^®^) identified numerous substitutions, some of which are shared across multiple genotypes and sub-lineages in both analyses. This raises questions about how these changes may affect vaccine-induced immunity and vaccine efficacy. Overall, this study highlights the critical need to sequence and analyze the genetic evolution of DENV in order to develop more effective prevention and control strategies, as well as to define vaccines best suited to circulating strains.

## Figures and Tables

**Figure 1 vaccines-13-00264-f001:**
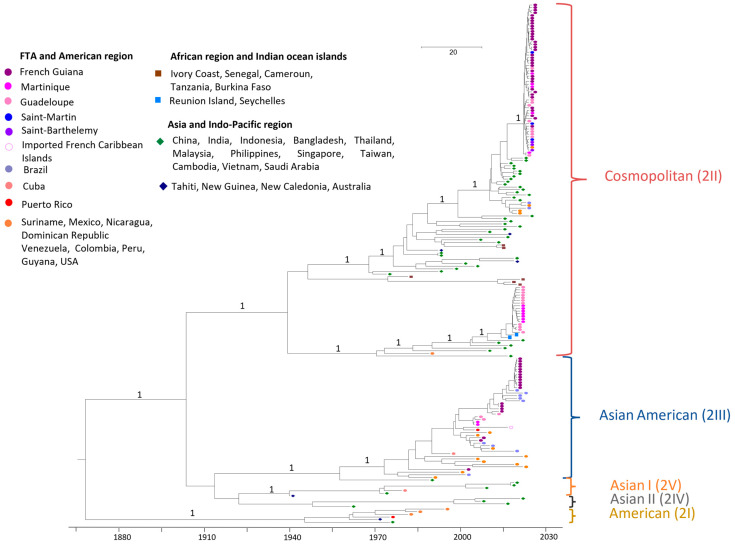
Bayesian phylogeny of DENV2 whole-genome sequences. The maximum clade credibility (MCC) tree, using GTR + G + I (General Time Reversible with Gamma distribution and Invariant sites) substitution model, with strict lognormal clock model and Bayesian skyline prior. The analysis included 199 sequences with 77 FTA sequences generated in this study and 122 reference sequences downloaded from GenBank. Sylvatic strains (OK605757 and EF105384) were used as the outgroup. The new classification proposed by Hill et al. has been added in brackets [26].

**Figure 2 vaccines-13-00264-f002:**
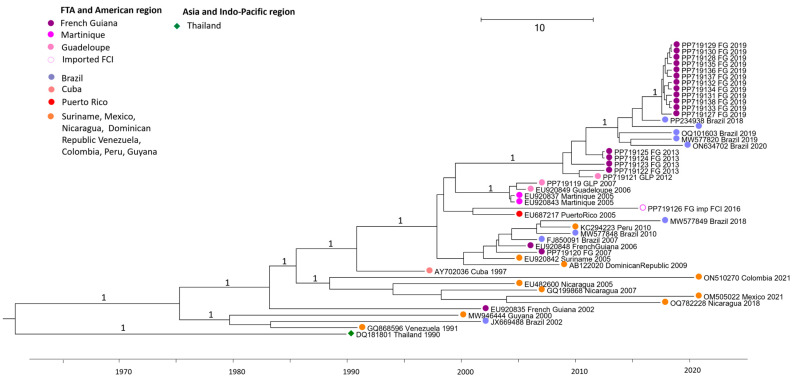
Bayesian phylogeny of DENV2 whole-genome sequences. Detailed on Asian American genotype. The maximum clade credibility (MCC) tree, using the GTR + G + I (General Time Reversible with Gamma distribution and Invariant sites) substitution model, was constructed with a strict clock model and Bayesian skyline prior.

**Figure 3 vaccines-13-00264-f003:**
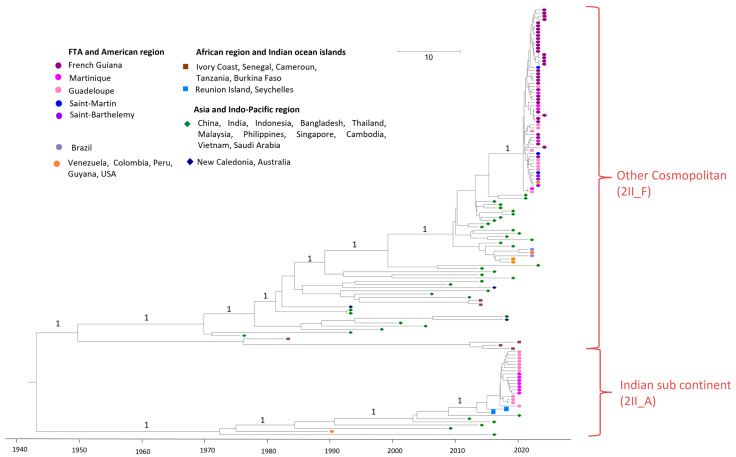
Bayesian phylogeny of DENV2 whole-genome sequences. Detailed on Cosmopolitan genotype. The maximum clade credibility (MCC) tree, using the GTR + G + I (General Time Reversible with Gamma distribution and Invariant sites) substitution model, was constructed with a strict clock model and Bayesian skyline prior. The new classification proposed by Hill et al. has been added in brackets [26].

**Figure 4 vaccines-13-00264-f004:**
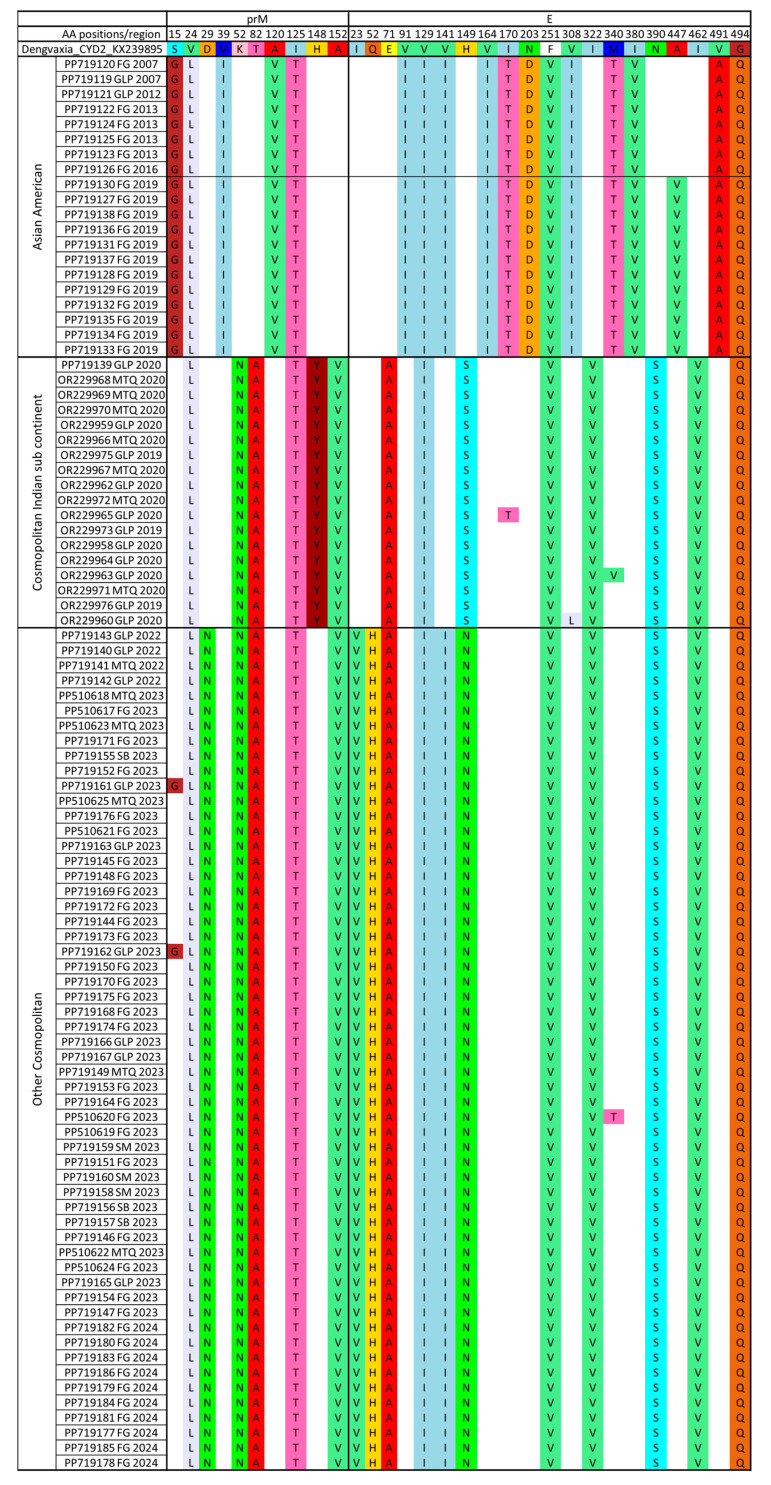
Mapping of observed AA substitution patterns specific to one or more genotypes and sub genotypes (Asian American, Cosmopolitan Indian Sub-continent, and Cosmopolitan Other Cosmopolitan) compared with the Dengvaxia CYD-2 vaccine strain (KX239895). The A447V AA change is observed for a sublineage of the Asian American genotype.

**Figure 5 vaccines-13-00264-f005:**
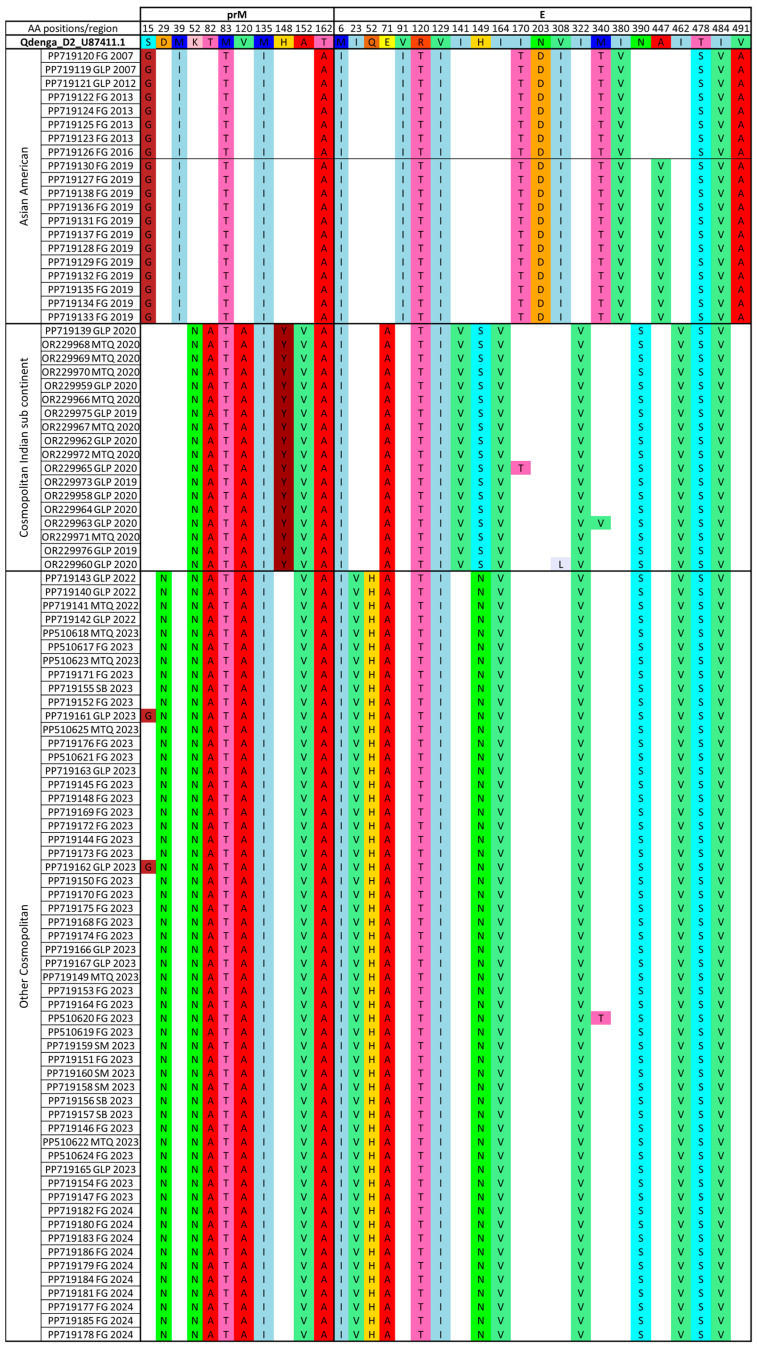
Mapping of observed AA substitution patterns specific to one or more genotypes and subgenotypes (Asian American, Cosmopolitan Indian Sub-continent, and Cosmopolitan Other Cosmopolitan) compared with the Qdenga vaccine strain (U87411). The A447V AA change is observed for a sublineage of the Asian American genotype.

**Table 1 vaccines-13-00264-t001:** Number of generated sequences by territory/municipalities and years of sampling.

Territory	n	Municipality	Year
Guadeloupe	**12**		
	1	Baie-Mahault	2022
	1	Basse-Terre	2007
	1	Les Abymes	2012
	2	Pointe-à-Pitre	2022
	1	Sainte-Anne	2020
	6	Uninformed	2023
Martinique	**5**		
	4	Fort-de-France	2023
	1	Lamentin	2022
Saint-Barthelemy	**3**	Uninformed	2023
Saint-Martin	**3**	Uninformed	2023
French Guiana	**51**		
	9	Cayenne	2007/2013/2019/2023
	14	Kourou	2013/2019/2023/2024
	12	Remire-Montjoly	2013/2023/2024
	5	Matoury	2023/2024
	3	Tonate-Macouria	2019/2023
	2	Sinnamary	2023/2024
	2	St-Laurent-du-Maroni	2013/2023
	2	Montsinery-Tonnegrande	2023/2024
	1	Saint-Georges	2023
	1	Roura	2024
French Guiana imported from French Caribbean Islands	**3**		
	1	Guadeloupe	2023
	1	Martinique	2023
	1	Uninformed	2016
**Total**	**77**		

**Table 2 vaccines-13-00264-t002:** AA substitution found in both analyses (Dengvaxia and Qdenga vaccine strains) in the prM and E regions.

Region prM	n = 8	S15G, D29N, M39I, K52N, T82A, A120V or V120A, H148Y, A152V
Region E	n = 18	I23V, Q52H, E71A, V91I, V129I, I141V or V141I, H149S and N, I164V or V164I, I170T, N203D, V308I, I322V, M340T, I380V, N390S, A447V, I462V, V491A

## Data Availability

All virus sequences are accessible on Genbank (accessions specified in the Appendix A). The Python code and input files are available on Github at https://github.com/aliselagrave/Python-script-to-compare-DENV-vaccine-strains-and-a-sequence-alignment-of-DENV2-.git (accessed on 14 December 2024).

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
