# Peer review of "The Genetic Evolution of DENV2 in the French Territories of the Americas: A Retrospective Study from the 2000s to the 2024 Epidemic, Including a Comparison of Amino Acid Changes with Vaccine Strains"

_vaccines, 2025, doi:10.3390/vaccines13030264_

Round 1

Reviewer 1 Report

Comments and Suggestions for Authors

Some editing is required locations and immune 

Significance of Hyperendemic?

Selection 'of'' 77 samples...l104

Complementarity...l112

Etc

p8

Reports on the composition of the serum samples and at what age of the viral infection they were collected

Complementary...    l112 etc

the status of the infected donors

Ans was replication still very active

Reviewer 2 Report

Comments and Suggestions for Authors

In this manuscript, the authors study the genetic diversity and dynamic of dengue virus (DENV) in French territories of the Americas (FTA) and try to correlate these with the possible effect on two commercial vaccines available to date, Dengvaxia and Qdenga. In particular, the authors add whole genome sequences of DENV-2 (N=77) from French Guiana and other regions and use them in the analysis.

This is an excellent study focusing on the DENV2 genetic diversity and its possible effect on immune evasion and vaccine efficacy. The manuscript is easy to follow and supported by rigorous methods and clearly presented results. This reviewer thinks that the authors are wise enough not to force the conclusion to make a direct correlation between genetic diversity and vaccine efficacy or immune evasion since multiple factors are involved and not covered by the current study.

I have few comments/suggestions:

1.      Samples selection: Apart from the low Ct value to ensure the sequencing success, what are other selection criteria used in selecting the samples for sequencing? Do the authors also try to incorporate the different clinical manifestations of the patients?

2.      Are there clinical data available to correlate with the sequence data? It will be interesting to correlate the disease severity/clinical manifestation or infection status (primary or secondary) with the sequence data.

3.      Amino acid changes were observed when comparing the 77 DENV-2 sequences with the other 17 sequences from Martinique and Guadeloupe and Dengvaxia and Qdengue sequences. This is somehow expected because DENV evolves all the time. In addition, the original strain of Dengvaxia is the Thailand DENV-2 PUO-218 isolated in 1980, while DENV-2 Thailand 16681 used in Qdenga was isolated in 1984; both are of Asian I genotype (Martinez DR, et al. Cell Rep 2020 and Phadungsombat J, et al. PloS ONE 2018). Might be useful to add this information on the distinct genotypes used in vaccines and genotypes currently circulating.

4.      The authors mention that the AA changes were also previously identified by Martinez et al. from a panel of DENV2 sequences and these AAs may correlate with immune evasion. Do the amino acid changes found in known epitopes in this study (Lines 325-326) correlate to positions of mAbs targeting, as found in another published report (Velasco JM, et al. PloS NTD 2024)? Might be an additional discussion.

5.      I agree with the conclusion that the obtained results raise questions about how genetic changes may affect vaccine-induced immunity and vaccine efficacy. The gold standard to do this is the PRNT method, which analyses antibodies from serum samples. It is recommended to add this to the discussion.

Minor comments:

1.      Describe the method to do the RNA extraction

2.      If the primer sequences listed in the Appendix Table are already published elsewhere (Quick et al), no need to include this again in the manuscript.

Reviewer 3 Report

Comments and Suggestions for Authors

The paper describes the genomic sequences of 199 Dengue 2 strains isolated in the French territories of America in the period 200-2024. The phylogenetic analysis shows that up to 2019 the region was populated by Asian American clades which were then replaced by Cosmopolitan clades which are likely to be imported from Indian regions. The new clades were associated with an important outbreak. The recently circulating strains are shown to have many mutations in the envelope proteins of the virus compared to the licensed vaccine strains suggesting that escape from vaccine- induced neutralizing antibodies is likely to occur.

Overall, the paper is well done and worth publishing.

Minor comments

The paper would gain a lot of value by adding data on the on how vaccine-induced antibodies neutralize the recent strains.

A figure showing where the mutations map on the structure of the envelop proteins would help to better understand the message of the paper.
